# Calcareous Nannofossils Biostratigraphy of Late Cretaceous–Paleocene Successions from Northern Jordan and Their Implications for Basin Analysis

**Albesher Hussein, Osama M. Al-Tarawneh * and Mohammad Alqudah**

Department of Earth and Environmental Sciences, Faculty of Science, Yarmouk University, Shafeeq Irshedat Street, Irbid 211163, Jordan; albesherhussean@gmail.com (A.H.); mohammad.alqudah@yu.edu.jo (M.A.)
* Correspondence: 2018330010@alumni.yu.edu.jo; Tel.: +(962)777-085-223

**Abstract:** Local geological and tectonic processes have been pivotal in shaping the diverse sedimentation patterns observed in Jordan, forming sub-basins characterized by elevated organic matter content (TOC). This study aims to characterize the Maastrichtian basin, focusing on sedimentation rates using calcareous nannofossils and understanding paleoecological and paleo-oceanic conditions. It offers insights into the paleoenvironmental factors impacting oil shale deposition in the late Maastrichtian–Paleocene period. It employs classical biostratigraphical, semi-quantitative, and statistical methodologies to achieve its objectives of age determination and paleoecological insights. A total of 116 smear slides from two sites were obtained: the first, consisting of WA-1 (23 samples), WA-2 (18 samples), and WA-3 (11 samples), and the second, with 60 samples. Notably, the sites exhibit varying topography. WA-1 and WA-2, situated at lower elevations, have the highest Total Organic Carbon (TOC) levels, while areas with higher elevations in section four are visually identified by a light color. The study revealed varying patterns of calcareous nannofossil richness in the two investigated sites. These patterns were instrumental in defining biozones, with the utilization of marker species such as *Lithraphidites quadratus*, *Micula murus*, *Micula prinsii*, and *Cruciplacolithus tenuis*. Chronologically, these sections were classified as Maastrichtian–Paleogene, encompassing the following biozones in sequential order: UC-20a, UC-20b, UC-20c, UC-20d, and NP-2. Furthermore, the study identified two hiatus intervals, observed in sections WA-1 and KAS-1. The absence of certain biozones in the analyzed sections suggests that these sections correspond to distinct geological blocks within the basin, underscoring the role of tectonic forces during the deposition period. The sedimentation rate initially commenced at low levels but gradually increased due to topographic alterations. Notably, the biozone UC-20c demonstrated a clear trend toward warming and enhanced nutrient availability. In this context, the abundance and diversity of species were associated with increased continental influx into the sub-basin, resulting in rising nutrient levels and the number of calcareous nannofossils. This study enhances the understanding of the local and global effects such as tectonic and climates of the continuity of basins by deciphering calcareous nannofossil patterns and their correlation with sedimentation factors.

**Keywords:** biostratigraphy; abundances; diversity; terrigenous influx

## 1. Introduction

In the realm of marine life, calcareous nannoplanktons have played a crucial role as primary producers in the open ocean for the past 250 million years [1]. Calcareous nannofossils are organisms smaller than 30 microns, unicellular in nature, with golden pigments, and they construct their protective shells from calcium carbonate [2]. An important role played by calcareous nannofossils in basin analysis is in unraveling the geological history and environmental dynamics of sedimentary basins. Their high sensitivity makes them reliable biostratigraphic markers, enabling the construction of chronological frameworks [3].

Furthermore, these microfossils can serve as paleoenvironmental indicators, reflecting temperature, nutrient availability, and water column structure [1]. They also contribute to our understanding of paleoceanography by revealing past oceanographic conditions. Moreover, they can be used to decipher sedimentation patterns, identify hiatuses, and assess sediment provenance [4]. Their role in hydrocarbon exploration is to provide insights into the quality of source rocks and reservoirs [1]. In addition to providing insight into past climate change events, these microfossils also provide insight into climate events [4]. The importance of sedimentary basins is underscored by their multifaceted role in deciphering Earth's history. The Muwaqqar Chalk Marl Formation (MCM), situated in the eastern regions of the Jordanian plateau, extends over expansive areas encompassing the Risha area, the Desert Highway, and northern Jordan [5–7]. This geological formation's lower section is notable for its rich oil shale deposits in Jordan and comprises a consistent, soft chalk-marl material throughout its thickness. The (MCM) is dated to the Maastrichtian–Paleocene age [8,9]. During the MCM's deposition, the Jordanian and adjacent regions underwent a significant marine transgression, transitioning them into the outer continental shelf, as elucidated by the authors of reference [10]. The presence of an abundance of planktonic foraminifera in northern Jordan provides strong evidence of a pelagic environment during the formation of the MCM [8]. Moreover, there are indications that the segment of the formation housing the oil shale deposits was influenced by cold, deep upwelling currents originating from the Neo-Tethys, which enriched the photic zone's water with essential nutrients and organic matter, thereby enhancing biological productivity [11,12].

In a study by Alqudah [13], they observed a high abundance of reworked Cretaceous and Paleocene calcareous nannofossils in samples of oil shale from five wells across Jordan. This abundance was suggested to be linked to active graben flank movements, which increased accommodation space in Eocene basins. In the same year, Farouk [14] used planktonic foraminifera and calcareous nannofossils to identify the Cretaceous/Paleogene boundary in Jordan. Their research indicated a hiatus extending from the late Maastrichtian to the early Danian stages. They also noted a transgression that led to deep-water conditions (Zones P4 or equivalent NP7/8), resulting in the deposition of a retrogradation parasequence set of middle-shelf pelagic marl and chalk during a rapid rise in sea level. Additionally, Gomez [15] combined prior findings of clay mineralogy, geochemistry, and benthic foraminifera with new insights into sulfur content and calcareous nannofossils. They found that variations in the organic matter content were primarily influenced by nutrient fluxes. Previous investigations had dated the lower part of this section to the Eocene Epoch in the Wadi Shallala Formation in northwestern Jordan [16]. Chronostratigraphically, the formational boundary between the Wadi Shallala and the Umm Rijam Chert Limestone in Wadi Shallala indicated regional paleoenvironmental changes in Jordan, which were closely correlated with neighboring countries like Egypt. In a study conducted by Alhejoj [17], they examined foraminiferal assemblages from the upper Maastrichtian to the middle Eocene succession in Jabal Ghuzayma. Using qualitative and quantitative analyses, they identified nine planktonic biozones spanning the upper Maastrichtian to the middle Eocene, along with two significant time gaps between the Paleocene/Eocene (P/E) and Cretaceous/Paleogene (K/Pg) unconformities.

This is prompted by the growing interest in oil shale as an energy source, primarily attributed to its origin from the biological productivity and preservation of organic material from deceased organisms. Indeed, research in the last two decades has increasingly focused on discerning the age and environmental context of oil shale deposition. However, concerns have arisen regarding the continuity of oil shale deposits within the basin. This variability in the presence of bitumen limestone throughout the basin is influenced by the effects of ancient landscape features (paleotopography). A central objective of this study is to gain a better understanding of the conditions that shaped the basin during the deposition period. This study aims to (a) provide a detailed characterization of the Maastrichtian basin, including its various components, with a particular emphasis on understanding sedimentation rates through an analysis of calcareous nannofossil assemblages; and (b) define

the paleoecological and paleo-oceanic conditions of the Maastrichtian epicontinental sea. These insights can shed light on the paleoenvironmental processes that influenced oil shale deposition in the basin during the late Maastrichtian–Paleocene period.

## 2. Study Area

### 2.1. Location Area and Geological Setting

Two specific sites within northern Jordan were chosen for investigating the calcareous nannofossil biostratigraphy assemblages. The initial site is situated in Wadi Al-Arab, encompassing sections WA-1, WA-2, and WA-3, with coordinates at 32°36′33.16″ N and 35°39′26.53″ E. This area includes the valley that directly feeds into the Wadi Al-Arab Dam. The second site is positioned along the path connecting Kufr Asad and Sifin, identified as section KAS, and can be located at 32°37′33.31″ N and 35°43′7.86″ E, as depicted in Figure 1.

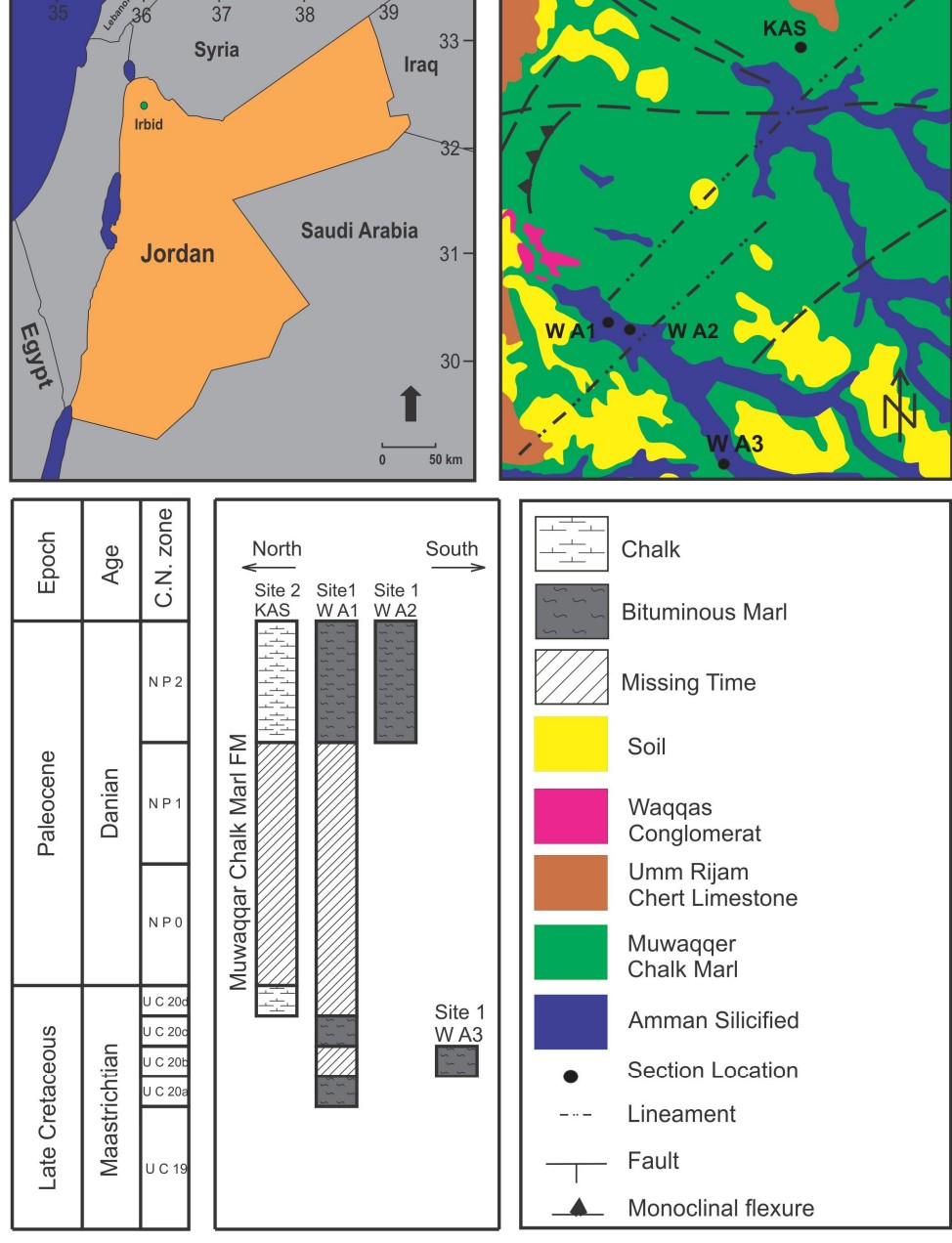

**Figure 1.** Location and geology map of the study area. The geological time scale adopted is the recent scale [18]; the source of the geological map [19].

The collected samples originated from the MCM Formation, which is widely exposed in various regions of Jordan. These areas include the eastern portion of the Jordanian plateau, the Risha area, the eastern sector of the Desert Highway, and the far northern regions of Jordan [6,7,20]. During the Late Cretaceous period, Jordan was situated on a broad and shallow shelf along the southern edge of the Neo-Tethys Ocean, a geographic context confirmed by the authors of references [6,10,21,22].

From a sedimentological perspective, The MCM formation, known for its hosting of Jordanian oil shale, predominantly comprises substantial layers of chalky marls, soft chalks, marls, marly limestones, and occasionally microcrystalline limestones [6,8,23,24]. The Late Cretaceous shelf in the northern part of the African–Arabian Plate was significantly influenced by tectonic movements, specifically the convergence of the Eurasian Plate toward the African–Arabian Plate and the Palestine Sinai Plate, as well as the presence of the Syrian Arc Fold belt [12,25]. Plate tectonic activities played a major role in the formation of basins during the Maastrichtian stage, operating on both regional and local scales [13,16,26].

*2.2. Biostratigraphy*

Biostratigraphy is a multi-step process as it involves species identification, followed by the division of the geological column into biostratigraphic zones. These zones are established based on the presence and prevalence of specific fossils, and they provide a means to track both the horizontal and vertical distribution of these fossils [27].

Burnett [28] introduced a biostratigraphic framework for the Upper Cretaceous (UC) period by analyzing data derived from Cretaceous records. Their approach combines bio-events sourced from different paleolatitudes and biogeographical regions, providing insights into marine paleoenvironments. This study aligns with the recent geological timescale [18].

The first and last occurrences (FO, LO) of calcareous nannofossil species have conventionally been regarded as synchronous worldwide [29]. However, numerous climatic shifts have heightened the bio-provinciality of calcareous nannoplankton assemblages. Notably, the Eocene stage is marked by a significant temperature disparity between the polar and equatorial regions [30,31]. Consequently, calcareous nannofossils appeared in relatively brief intervals, facilitating the establishment of robust biozonation frameworks based on these assemblages.

**3. Materials and Methods**

*3.1. Sampling*

This study focused on examining part of the MCM outcrop in the northern region of Jordan. A total of 116 samples were collected, and distributed across two distinct sites. The first site, Wadi-Arab, encompassed three sections, each located in different areas with varying levels of organic matter content (WA-1, WA-2, WA-3) (Figure 2). Specifically, twenty-four samples were collected from the first section, nineteen from the second, and thirteen from the third section. Conversely, along the Kufr Asad-Sifin road (KAS), a total of sixty-one samples were obtained from two continuous sections. The first section contributed ten samples, while the second section provided sixty-one samples for analysis.

The sampling approach in this study followed Jacob's method, where a standard 30 cm sample interval was maintained for all sections [32]. However, the first 20 samples from the Kufr Asad-Sifin area had a larger 1 m interval. This variation in sample interval was necessitated by the differences in the topography and slope of the various locations. All samples were extracted as core plugs to accurately document their original field positions. The determination of the sampling interval was initially guided by preliminary findings, with additional samples collected to facilitate high-resolution age analysis. Initially, the samples were primarily intended for a low-resolution study.

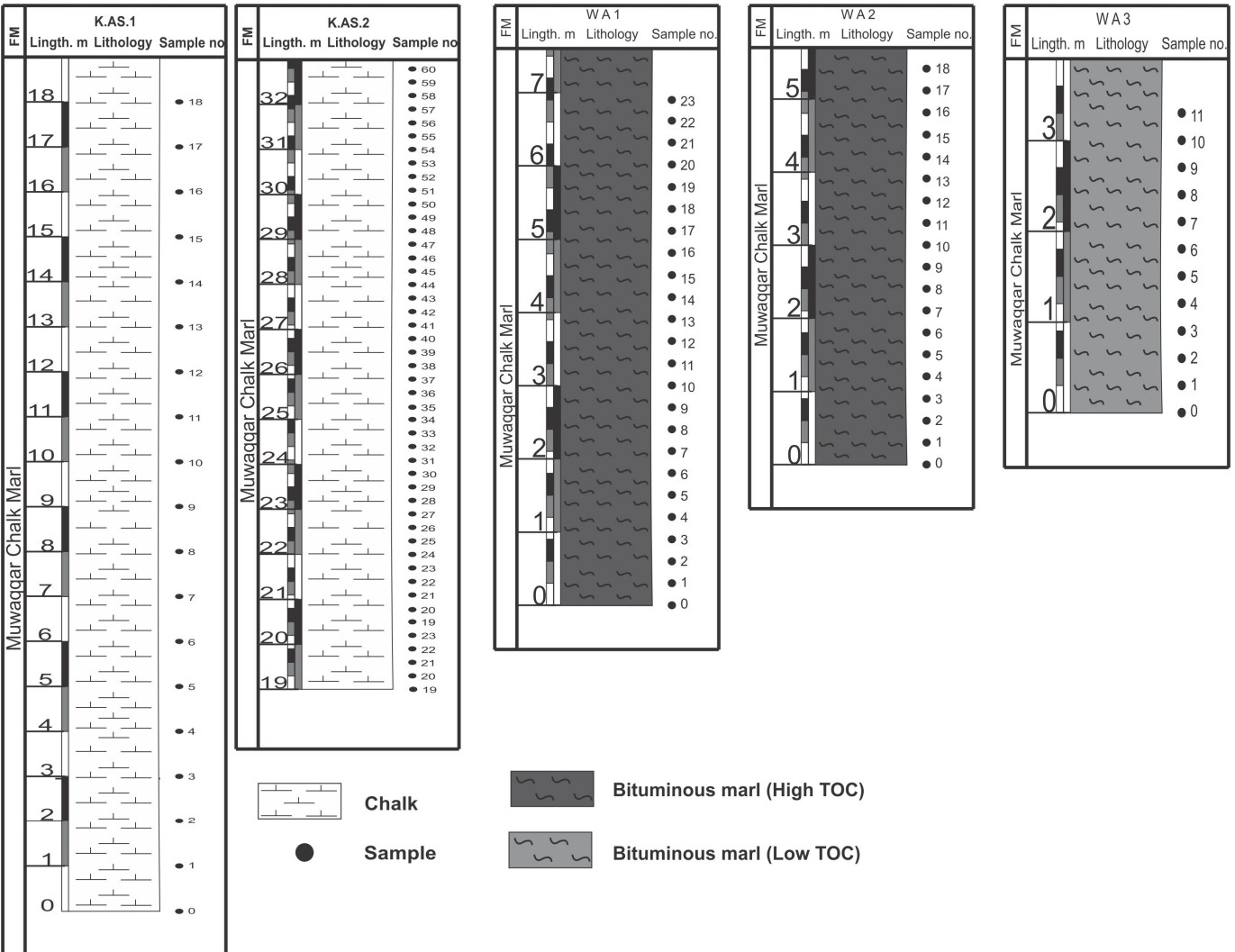

**Figure 2.** Lithology and position of samples in the different logs; the basin shows different topographies with different organic matter contents, according to the colors of rocks, by which the higher organic contents at the lowest topographies included two sections with a 30 cm sampling interval. The third section has a higher relief of less organic Carbone content with the same sampling interval as in the first two sections. In contrast, the highest topography that lacked the preservation of organic carbon has to interval affected by the high slope at the first 18 m, as the interval is 1 m and, later, the interval is retained at a 30 cm sampling interval.

### 3.2. Nannofossils Examinations

To study nannofossils, we followed the established procedure outlined in Bown and Young [2] for preparing simple smear slides. This process involved trimming the external surface of the rock samples, followed by careful scraping with a scalpel to ensure an even distribution of material on the slides. A flat-sided toothpick, along with a drop of distilled water, was then employed to spread the sediments thinly across the slide's surface. This allowed for a uniform suspension of the material, and each sample was identified separately. Subsequently, the slides were allowed to air-dry as the final step before they were covered and prepared for analysis.

Calcareous nannofossils were investigated through the utilization of a polarized microscope with a magnification set at 1500×. Each smear slide was meticulously examined, and within each field of view, specimens were identified, documented, and quantified following the taxonomic criteria [2]. The samples encompassed a time frame ranging

from the Maastrichtian to the early Paleogene. In every field of view, a minimum of 300 specimens was counted.

The biostratigraphy scheme and the assessment of calcareous nannofossil relative abundance were conducted based on smear slides created [2]. The uniform distribution achieved during slide preparation not only simplifies the counting process but also aids in the dispersion of calcareous nannofossils, thus enhancing their examination for specimen identifications [33].

### 3.2.1. Taxonomy

The taxonomic classification of the species was established based on the biozonation schemes, which cover the Cretaceous and Paleocene periods [2,28,34]. Burnett [28] employed the abbreviation "UC" to indicate the Upper Cretaceous in their biozonation scheme. All recognized taxa are comprehensively documented in the systematic paleontology section and are systematically listed in the taxonomic table for reference.

### 3.2.2. Preservation/Dissolution Indices

By the preservation stages, the extent of preservation has been quantified through the assessment of etching (E) and overgrowth (O). Preservation levels are denoted as follows: E1 and O1 correspond to slight preservation, E2 and O2 indicate moderate preservation, and E3 and O3 signify poor preservation [3,35].

### 3.2.3. Statistical Analysis

The Shannon diversity index (H) was employed, and its values are influenced by the level of heterogeneity [36]. This index primarily factors in both the number of taxa and the number of individuals. The Shannon diversity index equation is as follows:

$$H = -\Sigma \, (ni/n) \, \ln(ni/n) \qquad (1)$$

In this formula, n is the total number of individuals, and ni is the number of individuals of taxon i.

### 3.2.4. Calcareous Nannofossil Nutrient and Temperature Indices

Calcareous nannofossils are affected by oceanographic and climatic conditions. Consequently, these fossils provide a reconstruction method to display the variations in the paleoenvironmental conditions throughout geological time [37]. Additionally, nutrients and surface water temperature play a major role in investigating the abundance and distribution of calcareous nannofossils. Therefore, fluctuations in surface temperature and nutrient levels were assessed using a nannofossil-based nutrient and temperature index.

In this study, we employed the Low Nutrient Index (LNI) and Cold-Water Index (CWI) to unveil prevailing nutrient and sea-surface temperature conditions during the specified time intervals. Our approach aligns with the methodology proposed by Aizawa [38].

The relations are as follows:

$$LNI = \text{Low nutrient taxa}/(\text{High Nutrient taxa} + \text{Low nutrient taxa}) \times 100 \qquad (2)$$

And the (CWI) as follows:

$$CWI = \text{Cold taxa}/(\text{Warm taxa} + \text{Cold taxa}) \times 100 \qquad (3)$$

### 3.2.5. Sedimentation Rate

We used a specialized approach to the utility of microfossils in paleostratigraphic and basin architecture analysis [39]. One of the key techniques within this framework is the graphical correlation technique, which employs a semi-quantitative method utilizing nannofossils. In this method, the vertical position of the initial and terminal occurrences of species is plotted on a graph [39].

The resulting line, referred to as the Line of Correlation (LOC), is employed to estimate sedimentation rates. The LOC is derived by assessing the relative sedimentation rate, with an upward trend indicating higher sedimentation rates, following the methodology [40]. The first and last occurrences of marker species in the Wadi al Arab and Sifin sections are plotted against those in the standard composite reference section of QC-28 [26].

## 4. Results

Calcareous nannofossil taxonomy encompasses data related to marker species, their abundance, and preservation. These raw data are instrumental in estimating the geological age of the stratum. The examination focused on calcareous nannofossils from chalk successions. Given that certain species' abundance is known to be influenced by environmental conditions, in which they serve as effective indicators of paleoenvironments, this study particularly emphasizes two categories of species: those that provide age-related information and those that indicate paleoenvironmental conditions. Furthermore, it emphasizes the appearance and disappearance of calcareous nannofossils concerning calcareous nannofossil horizons [40].

### 4.1. Calcareous Nannofossil Assemblages

A total of 17 calcareous nannofossil groups were identified in the samples, with 14 belonging to the Cretaceous period, including *Arkhangelskiella*, *Ceratolithoides*, *Cribrosphaerella*, *Eiffellithus*, *Gartnerago*, *Kamptnerius*, *Lithraphidites*, *Micula*, *Nephrolithus*, *Prediscosphaera*, *Reinhardtites*, *Tranolithus*, *Watznaueria*, and *Uniplanarius*. Furthermore, one group, *Biantholilhas*, which is commonly found in the Cretaceous/Paleogene Boundary and Early Paleocene, was also identified, along with two groups typical of the Paleocene, namely *Coccolithus* and *Cruciplacolithus*.

Marker species indicative of the Maastrichtian age, including *T. orionatus*, *L. quadratus*, *M. prinsii*, and *M. murus*, were identified as Bown [2] (see Figure 3). The presence of these species coincided with the absence of Campanian marker species, *Brionsonia parca parca* and *E. eximus*, as well as the Paleocene species *C. tenuis*, which signifies the early Paleocene age. These species, serving as indicators of temperature and fertility in the surface water column, such as *A. cymbiformis*, *G. segmentatum*, *K. magnificus*, *L. carniolensis*, *M. decussata*, *N. frequens*, *P. cretacea*, *Thoracosphaera*, *W. barnesiae*, and *Zeugrhabdotus*, were found in varying abundances. In contrast, Paleocene species were rare, making them unsuitable for paleoenvironmental investigations.

### 4.2. Biostratigraphy

The MCM Formation has been established as correlating equivalently with the Upper Campanian nannofossil Zones of UC-20A-NP2 in the Tethyan realm [28], as demonstrated in Figure 4. In section 1, the distinctive Biozone UC-20a emerged at its base, which was primarily due to the presence of samples 1 and 2 associated with this biozone. Notably, *L. quadratus* was the dominant species within samples 1 and 2. The thickness of this biozone in Section 1 was measured at 60 cm. However, it is worth noting that Biozone UC-20a was absent in Sections 2–4.

The identification of the base of Biozone UC-20a was based on the appearance of *L. quadratus*, while the top of Biozone UC-20a coincided with the base of UC-20b. It should be noted that UC-20b was not observed in Section 1 but was consistently present in section 3 across all samples (samples 1–12). Additionally, Biozone UC-20c was identified through samples 3–17 in Section 1. The lower boundary of this biozone was determined by the presence of *C. kemptneri* species. In contrast, its upper boundary was marked by the presence of *M. prinsii*, serving as an indicative sign of the initiation of UC-20d. Notably, the onset of UC-20d was indicated by the presence of *M. prinsii* and continued until reaching an unconformity.

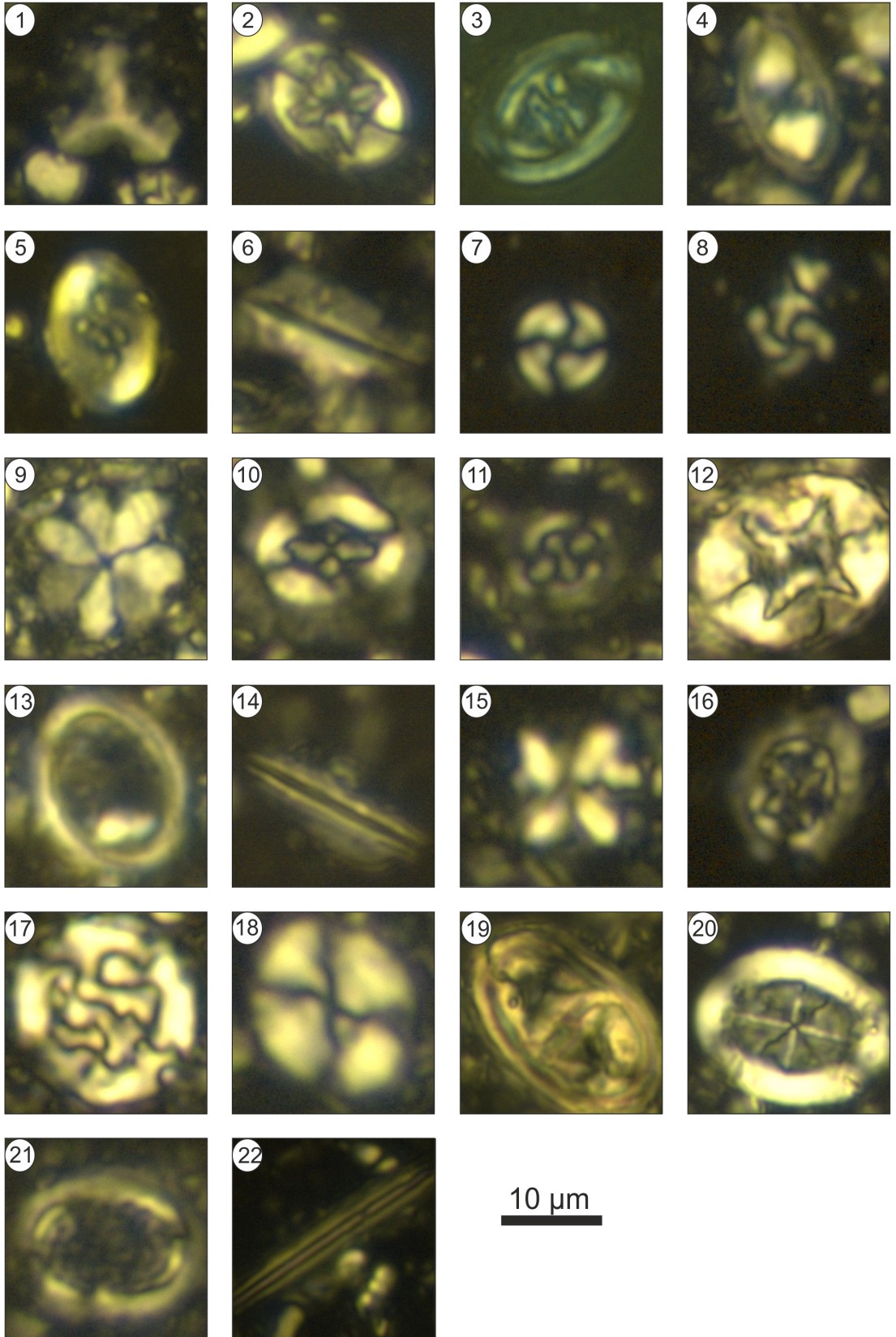

**Figure 3.** Main species discussed in this study. 1-*U.trifdus* (Stradner in stradner and Papps 1961) Prins and Perch-Nielsen in Manivit et al. 1977, 2-*E.* parallilus, Perch-Nielsen 1973 3-*R. anthophorus*, (Deflandre 1959) Perch-Nielsen 1968 4-*T. orionatus*, (Reinhardt 1966) Perch-Nielsen 1968 = *T. phacelosus* 5-*R. levis*, Prins and Sissingh in Sissingh 1977 6-*L. quadratus*, Bramlette and Martini 1964 7-*M. murus*,

(Martini 1961) Bukry 1973 8-*M. prinsii*, Perch-Nielsen 1979 9-*B. sparus*, Bramlette and Martini (1964) 10-*C. primus*, Perch-Nielsen 1977 11-*C. tenuis*, (Stradner 1961) Hay and Mohler in Hay et al. 1967 12-*E. turriseiffelii*, (Deflandre in Deflandre and Fert 1954) Reinhardt 1965 13-*G. segmentatum*, (Stover 1966) Thierstein 1974 14-*L. praequadratus*, Roth (1978) 15-*M. decussata*, Vekshina 1959 16-*P. cretacea*, (Arkhangelsky 1912) Gartner 1968 17-Thoracosphaera, Kamptner 1927 18-*W. barnesiae*, (Black 1959) Perch-Nielsen 1968 19-Zeugrhabdotus spp, Reinhardt (1965) 20-A. cymbiformis, Vekshina 1959 21-*C. ehrenbergii*, (Arkhangelsky 1912) Deflandre in Piveteau 1952 22-*C. pelagicus*, (Wallich 1871) Schiller 1930 22-*L. carniolensis* Deflandre 1963.

4.2.1. Biozonation

Biozone UC-20a was identified at the base of Section 1, as depicted in Figure 4, where samples 1 and 2 were attributed to this biozone. Notably, the dominant species in samples 1 and 2 was *L. quadratus*, and the thickness of this biozone in Section 1 measures 60 cm. However, Biozone UC-20a was conspicuously absent in Sections 2–4. The lower boundary of Biozone UC-20a was delineated based on the presence of *L. quadratus*, and its upper boundary coincided with the base of UC-20b. UC-20b was not observed in Section 1 but was consistently present in Section 3 across all 12 samples. In Section 1, samples 3–17 exhibited the occurrence of UC-20c. The lower boundary of UC-20c was remarked by the presence of *C. Kamptneri*, while its upper boundary was defined by *M. Prinsii*, marking the beginning of UC-20d. Indeed, UC-20d was identified by the appearance of *M. Prinsii*, persisting until an unconformity was encountered. This study adheres to the biozonations proposed in the conducted research, which encompassed an illustrative case study [26]. Consequently, a total of five distinct biozones have been delineated as follows:

*1-Lithraphidites quadratus* zone (UC20a), Age: Maastrichtian, Introduced by Bown and Young, 1998

Remarks: UC20a Zone is defined as the interval from the FO of *Lithraphidites quadratus* to the FO of *Micula murus*. This zone is recorded in samples 1 and 2 in Section 1, with a thickness around 0.6 m, and a hiatus at the top of UC20a.

*2-Micula murus* zone (UC20b), Age: Maastrichtian, Introduced by Bown and Young, 1998

Remarks: UC20b Zone is defined as the interval from the FO of *Micula murus* to the FO of *Ceratolithoides kamptneri*. This zone is recorded in samples 1–12 in Section 3, with a thickness of about 3.6 m.

*3-Ceratolithoides kamptneri* zone (UC20c), Age: Maastrichtian, Introduced by Bown and Young, 1998

Remarks: UC20c Zone is defined as the interval from the FO of *Ceratolithoides kamptneri* to the FO of *Micula prinsii*. This zone is recorded in samples 3–17 in Section 1, with a thickness of around 4.2 m, and a hiatus at the top of UC20c.

*4-Micula prinsii* zone (UC20d), Age: Maastrichtian, Introduced by Bown and Young, 1998

Remarks: UC20d Zone is defined as the interval from the FO of *Micula prinsii* to the LO of non-survivor Cretaceous taxa. This zone is recorded in samples 15–28 in Section 4, with a thickness of around 7.4 m, and a hiatus at the top of UC20d.

*5-Cruciplacolithus tenuis* zone (NP-2), Age: Danian, Introduced by Bown and Young, 1998

Remarks: NP2 Zone is defined as the interval from the FO of *Cruciplacolithus tenuis* to the FO of *Cruciplacolithus edwardsii*. This zone is recorded in samples 18–24 in Section 1 with a thickness of around (1.8 m), samples 7–19 in Section 2 with a thickness of around (3.6 m), and samples 29–61 in Section 4 with a thickness of around (9.6 m).

| Stage | Biozone | E. parallilus | R. anthophorus | T. orionatus | R. levus | L. quadratus | M. murus | C. kemptneri | M. prinsii | C. primus | B. sparus | C. tenuis | Section 1 Site 1 | Section 2 Site 1 | Section 3 Site 1 | Section 4 Site 2 |
|---|---|---|---|---|---|---|---|---|---|---|---|---|---|---|---|---|
| Danian | NP 2 | | | | | | | | | ■ | ■ | ■ | sample 18_24 | sample 7_19 | | sample 28_61 |
| Danian | NP 1 | | | | | | | | | | | | Gap | | | Gap |
| Danian | NP 0 | | | | | | | | | | | | Gap | | | Gap |
| Maastrichtian | UC 20d | ■ | | | | ■ | ■ | ■ | ■ | | | | Gap | | | sample 15_28 |
| Maastrichtian | UC 20c | ■ | | | | ■ | ■ | ■ | | | | | sample 3_17 | | | |
| Maastrichtian | UC 20b | ■ | | | | ■ | ■ | | | | | | Gap | sample 1_12 | | |
| Maastrichtian | UC 20a | ■ | | | | ■ | | | | | | | sample 1_2 | | | |
| Maastrichtian | UC 19 | ■ | | | ■ | | | | | | | | | | | |
| Maastrichtian | UC 18 | ■ | | ■ | | | | | | | | | | | | |
| Maastrichtian | UC 17 | ■ | ■ | | | | | | | | | | | | | |
| Maastrichtian | UC 16 | ■ | | | | | | | | | | | | | | |

**Figure 4.** The biostratigraphic scheme represents marker species and proposed biozonation.

4.2.2. Thicknesses and Preservation

The WA-1 section displayed predominantly moderate to well-preserved specimens along its length, with a notable absence of fossils at the terminus of the profile. Similarly, the WA-2 section demonstrated moderate to well-preserved specimens, yet both the initial and final portions of the profile were devoid of any biological remains. In contrast, the WS-3 section consistently displayed well-preserved samples throughout its length. Moreover, KAS exhibited fluctuating occurrences of marker species, with these taxa appearing and disappearing twice within the section. The preservation status was assessed based on the extent of etching and overgrowth, revealing a moderate degree of etching and overgrowth in the initial two sections, and a comparatively milder degree in the third section. The sections revealed varying thicknesses, with measurements of 6.5 m for WA-1, 5.5 m for WA-2, 3.5 m for WS-3, and 31 m for KAS.

*4.3. Continuity of Stratigraphic Sections*

In the WA-1 section, an elevated abundance of *Mecolla decussata* was observed during the UC20c interval, while other species exhibited low to extremely low occurrences (see Figure 4). It is important to note an evident unconformity at the end of UC20c, as the fossils discovered were reworked and not original to this layer. In contrast, the WA-2 samples under study showed no identifiable fossils; rather, the calcareous fossils present were reworked nannoplanktons, which were not expected to be part of the geological succession. Moving to the WA-3 section, a notable predominance of *M. decussata*, *W. barnesiae*, *P. cretacea*, and *E. turriseiffelii* was observed, while other detected fossils displayed significantly lower relative abundances. On the other hand, in the case of KAS, the samples showed a general increase in the relative abundance of *W. barnesiae* and *Zeugrhabdotus* spp. over geological time, while other species displayed fluctuations between rare occurrences and disappearances.

*4.4. Ecological Indicators*

Figures 5–8 provide graphical representations of the relative abundances of the identified species in the four sections. Notably, in section WA-1, *Micula decussata* exhibited the highest abundance during the UC20c interval, while *W. barnesiae*, *A. cymbiformis*, and *P. cretacea* had lower relative abundances. Conversely, *L. carniolensis*, *E. tur-riseiffelii*, and *Zeugrhabdotus* spp. displayed extremely low relative abundances. Furthermore, species such as *K. magnificus*, *L. praequadratus*, *E. parallellus*, *N. frequens*, *G. segmentatum*, and *C. ehrenbergii* were barely found. There is also a distinct unconformity noted at the end of UC-20c, as the studied species were identified as reworked fossils. The subsequent diagram illustrates each species in terms of their relative abundances (Figure 5).

The section WA-2 probed part revealed no remarked fossils in which the calcareous fossils were reworked nannoplanktons (Figure 6).

In section WA-3, both *M. decussata* and *W. barnesiae* demonstrated a notably high relative abundance, exceeding 125 specimens in some samples, as depicted in Figure 7. However, *M. decussata* displayed a sudden decrease in the last quarter, followed by a subsequent increase. Similarly, *A. cymbiformis*, *P. cretacea*, and *E. turriseiffelii* followed a nearly identical pattern, except for the final meter, which exhibited a distinct hiatus. *C. ehrenbergii* and *L. carniolensis* showed low relative abundance, with similar trends in the initial two-thirds of the section, followed by a reverse correlation in the final third, leading to a significant decline. *N. frequens* was absent from the majority of the section but exhibited two minor peaks in the middle. *Zeugrhabdotus* spp. displayed slightly higher relative abundance in the middle of the section as well.

In the KAS-1 section, there is a general trend of increasing relative abundance over geological time for *W. barnesiae* and *Zeugrhabdotus* spp. In contrast, *K. magnificus*, *N. frequens*, *C. ehrenbergii*, *M. decussata*, *L. carniolensis*, and *P. cretacea* generally exhibited a hiatus in the first two-thirds of the section and were represented by one or more peaks, such as those observed in *L. carniolensis* and *M. decussata*. *A. cymbiformis* displayed very low relative

abundance, while *L. praequadratus* was rarely found, and *G. segmentatum* was completely absent, as illustrated in Figure 8.

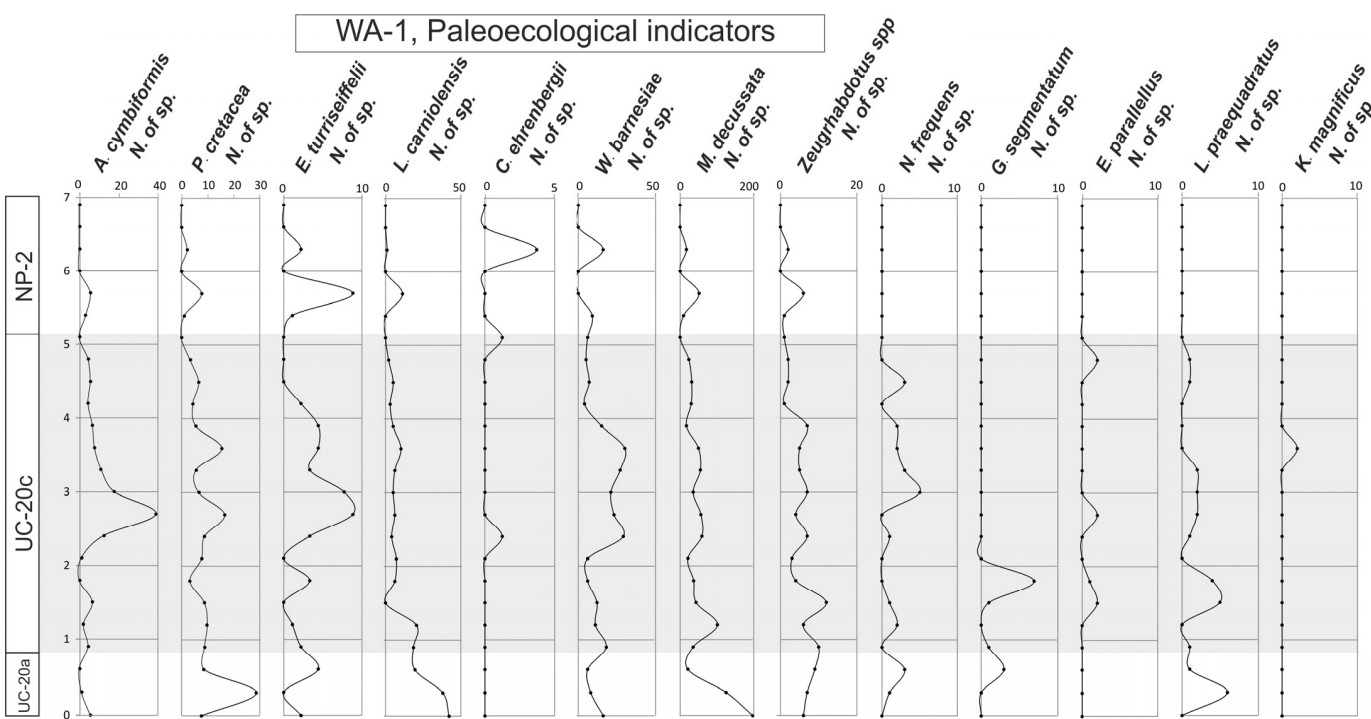

**Figure 5.** Relative abundances of calcareous nannofossil assemblages from WA-1.

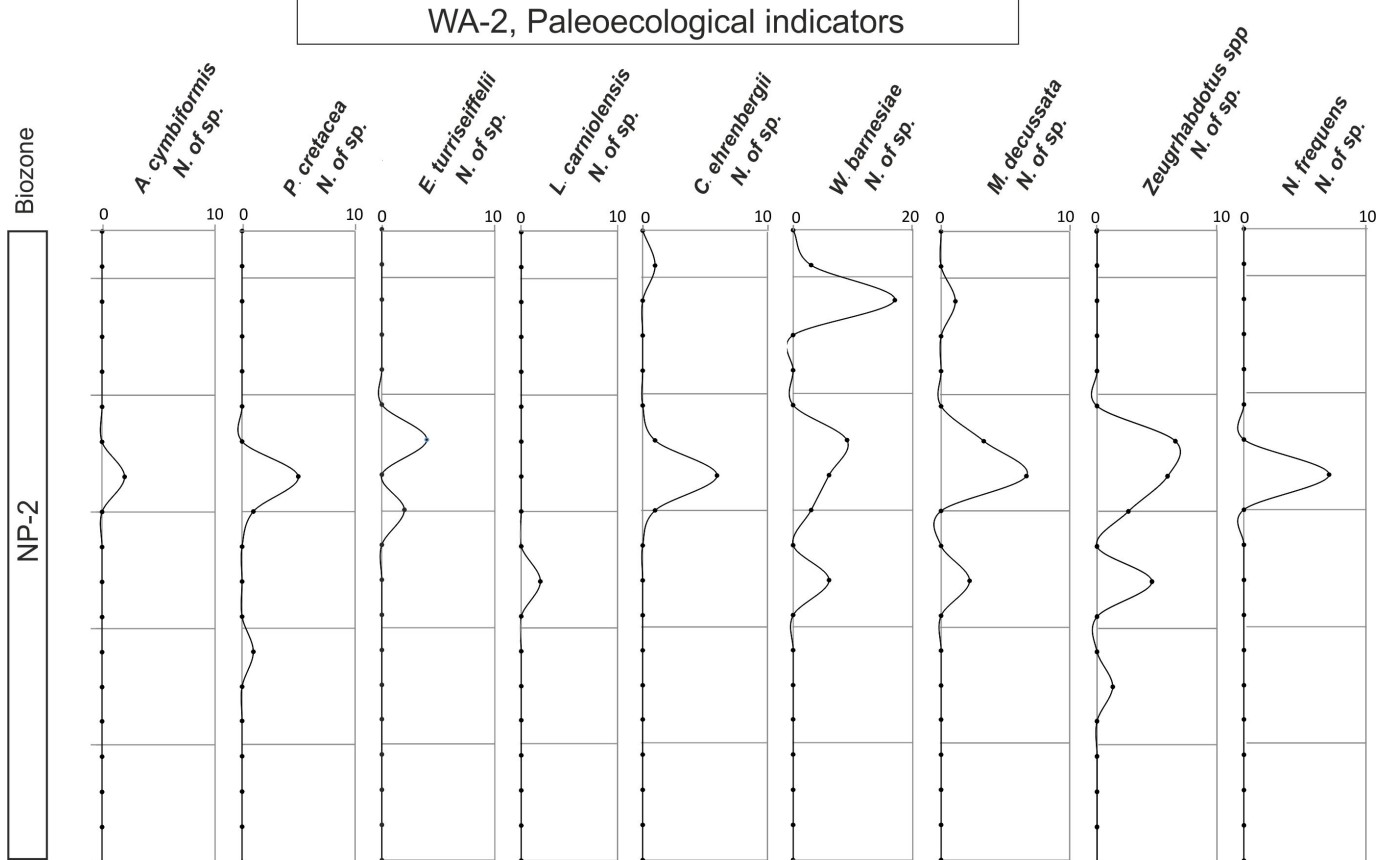

**Figure 6.** Relative abundances of calcareous nannofossil assemblages from WA-2.

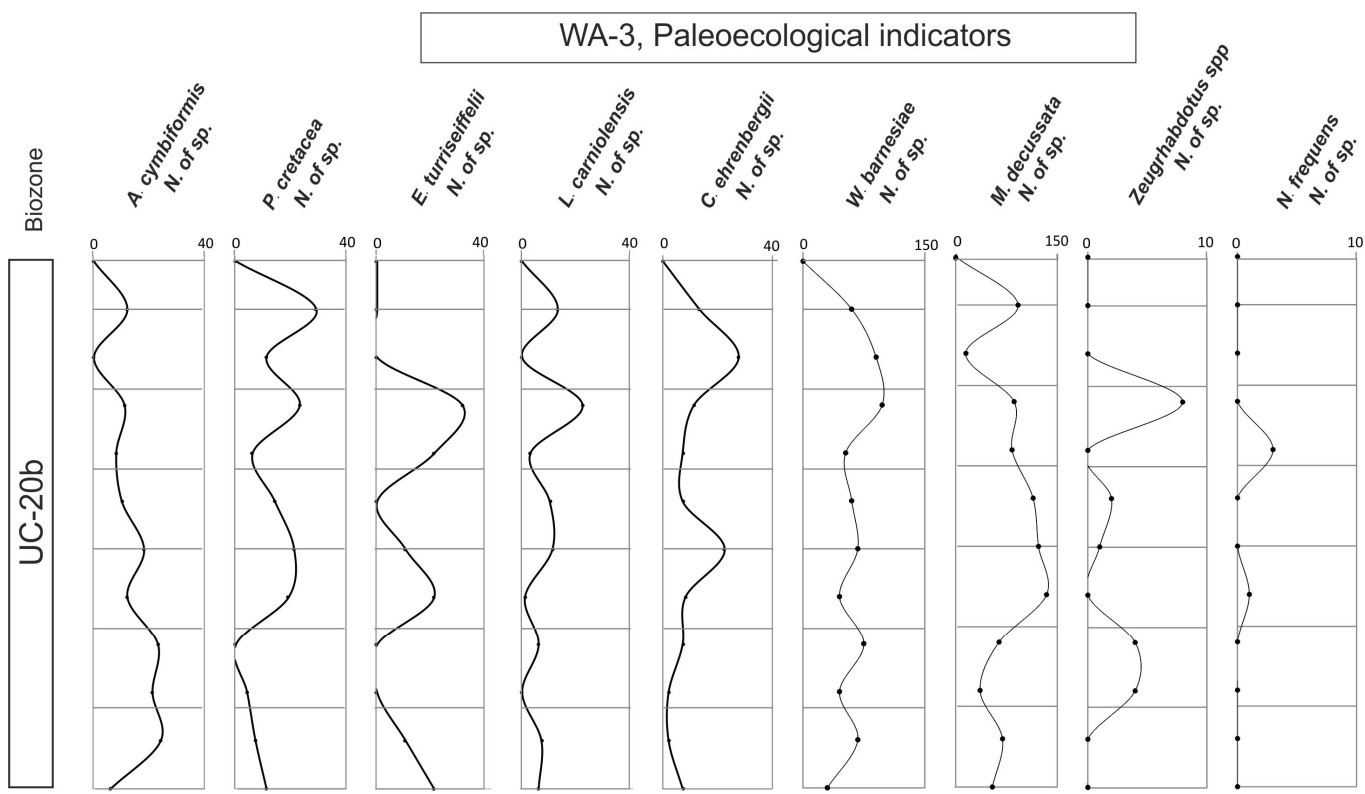

**Figure 7.** Relative abundances of calcareous nannofossil assemblages from WA-3.

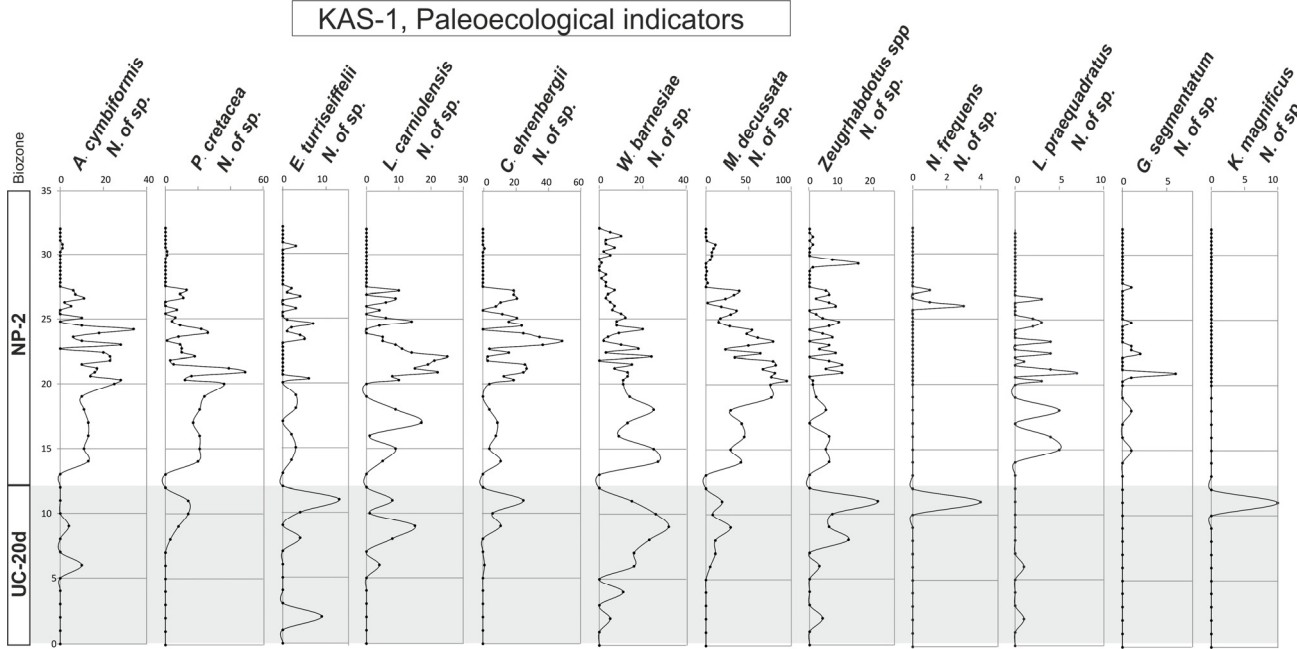

**Figure 8.** Relative abundances of calcareous nannofossil assemblages from KAS-1.

## 5. Discussion

### 5.1. Age Framework

The WA-1 section spans from biozone UC-20a to UC-20c, with a notable absence of biozone UC-20b, attributed to erosional processes triggered by rising sea levels, likely resulting from tectonic activity. While this absence was not previously evident in the northern part of Jordan, it had been documented in eastern basins [10,22,26]. Subsequently,

there is a gap in the sedimentary record extending into the Danian age until the biozone NP2, characterized by assemblages of reworked species [15,26]. The WA-2 section, a geographical extension of section one, chronologically falls within the biozone NP2, featuring assemblages suggesting Cretaceous age, likely due to the reworking of sediments. In contrast, the WA-3 section is situated within biozone UC-20b, recognized primarily based on the first occurrence [15]. Moving to the KAS-1 section, a hiatus is observed starting from the UC-20a, UC-20b, and UC-20c biozones, with the appearance of UC-20d marked by the presence of *M. prinsii* [14,17]. Another unconformity is noted as NP0 and NP1 are absent, and the successions show reworked species within the biozone NP2 [41].

### *5.2. Sedimentation Rates*

The graphical correlation of the resulting biostratigraphic data that have been obtained in this study is illustrated in Figure 9.

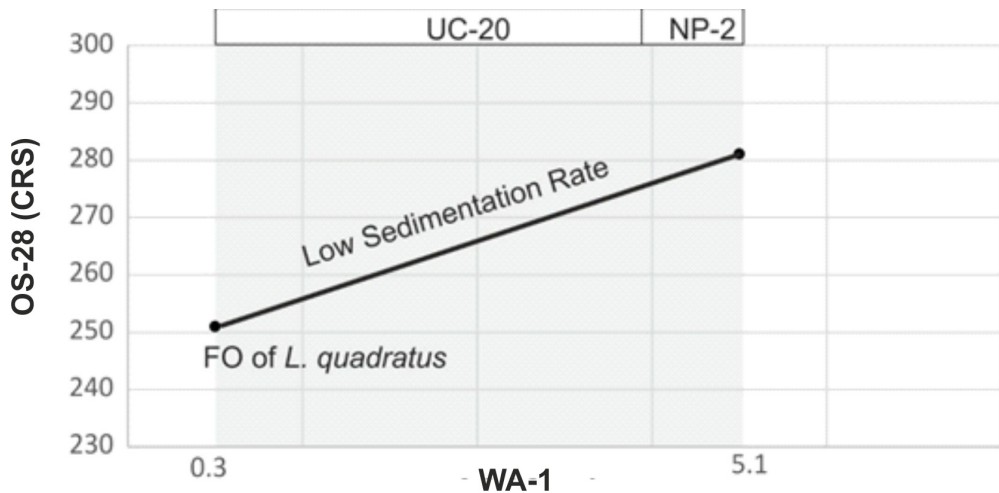

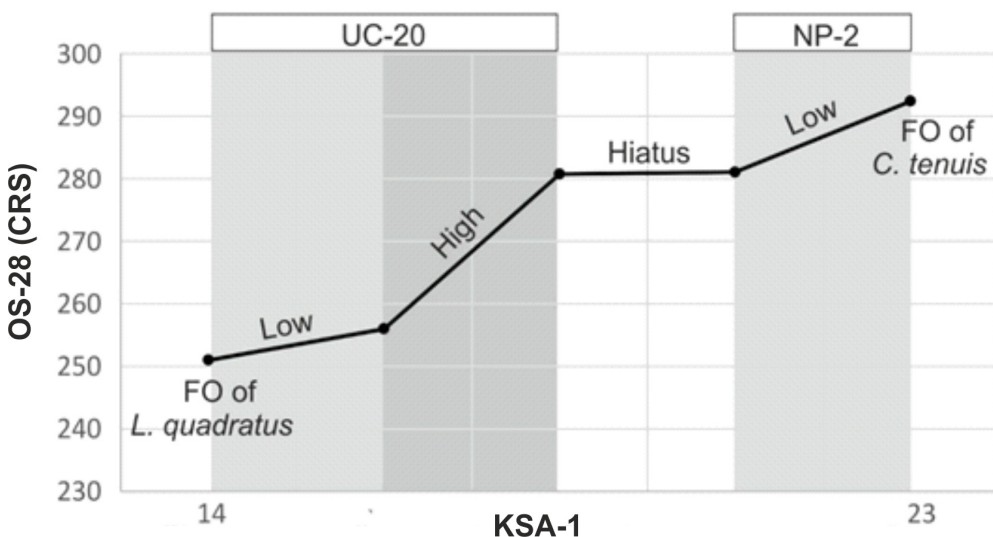

**Figure 9.** Line of correlation from two sections; WA-1 and KAS-1.

To establish a Line of Correlation (LOC), we plotted the stratigraphic positions of the last occurrences of *R. levis* and *T. orionatus*, as well as the first appearances of *L. quadratus*, *M. murus*, and *M. prinsii* in both sections WA-1 and KAS-1 against the composite standard reference section (OS-28) [26]. Section OS-28 has been chosen due to its comprehensive

stratigraphic representation spanning from the Late Cretaceous to the Eocene. The LOC provides insights into the relative sedimentation rates between the two compared sections [40]. Notably, both WA-1 and KAS-1 sections revealed a relatively low sedimentation rate during the early UC-20 interval.

The increase in sedimentation rate suggests an acceleration of basin subsidence and an elevation in topographic relief between geological blocks. The rise in sea level contributed to the establishment of a high sedimentation rate between the hinterland and the adjacent area [15,41]. Subsequently, a decline in sea level marked the end of this high-stand phase, initiating a period of hiatus during UC-20. This hiatus resulted in a regression of sedimentation, leading to noticeable gaps in the strata, as emphasized in the KAS section. The missing strata are evident in the Line of Correlation (LOC) as a plateau in Figure 9. Numerous studies by authors such as [15,24,26,42,43] have also reported the absence of strata within UC-20 and the early Paleocene.

### 5.3. Stratigraphy and Structural Model

Since the Late Cretaceous period, ongoing and significant tectonic activities have reshaped the basin, leading to dynamic changes in depositional environments and topography. The geological tectonic influence of the northern region is primarily governed by the tectonics of the Dead Sea transform fault.

Globally, during the late Cretaceous–Paleocene, the Earth experienced relatively high global sea levels. This was primarily due to a combination of factors, including a warmer climate, increased greenhouse gas concentrations, and the configuration of continents and ocean basins. Sea levels during this time were significantly higher than they are today, with many shallow epicontinental seas covering areas that are now land [44–46]. Locally, the Cretaceous–Paleocene periods show a rise in sea level, which is reflected in the rock types by which the deposition of the MCM. These rises were coupled with variations in the elevation and sedimentary influx enhancing the preservations in several sites along Jordan and neighboring countries [47–49].

Section WS-1 and WS-2 are situated within the same subbasin, with Section WS-2 exhibiting lower topographical elevation in comparison to Section WS-1 (refer to Figure 10). Both sections exhibited increased sedimentation rates and a higher organic matter content, as evidenced by the darker appearance of the samples. The hydrological cycle played a crucial role in enhancing sedimentation flux in the lower block, as sediments were conveyed from the neighboring block to the shoreline and extended into the distal sub-basin.

On the contrary, the reworking process of calcareous nannofossils is evident from the presence of Cretaceous assemblages in the Paleocene successions, including species like *A. cymbiformis*, *P. cretacea*, *E. turriseiffelii*, *L. carniolensis*, *C. ehrenbergii*, *W. barnesiae*, *M. decussata*, *Zeugrhabdotus* spp., and *N. frequens*. Section WS-3 is considered to be closer to the land and part of a larger fault system. It is characterized by generally low sedimentation rates and lighter-colored sediments, indicating lower organic matter content. In contrast, Section KSA-1 noted that the high topographical relief in this area led to the chalk stratum appearing devoid of organic matter.

### 5.4. Ecological Model and Oceanographic Implication

Calcareous nannofossils are influenced by climatic conditions, with sea surface temperature and nutrient levels being two crucial factors that shape the distribution of nannofossil assemblages [1,40]. The preferences of calcareous nannofossils, as illustrated in Figure 11 for sections WA-1, WA-3, and KAS, highlight certain species. In particular, *Arkhangelskiella cymbiformis*, *Watznaueria barnesiae*, *Meculla decussata*, *Zeugrhabdotus* spp., *Nephrolithus frequens*, *Gartnerago segmentatum*, and *Kamptnerius magnificus* typically serve as indicators of lower photic zones, signifying their presence in cold marine environments [1,15,29,40,50,51]. On the other hand, the Low Nutrient Index (LNI), based on preferences for species with low nutrient requirements, designates *P. cretacea*, *E. turriseiffelii*, *W. barnesiae*, and *M. decussata* as low fertility species [15,49].

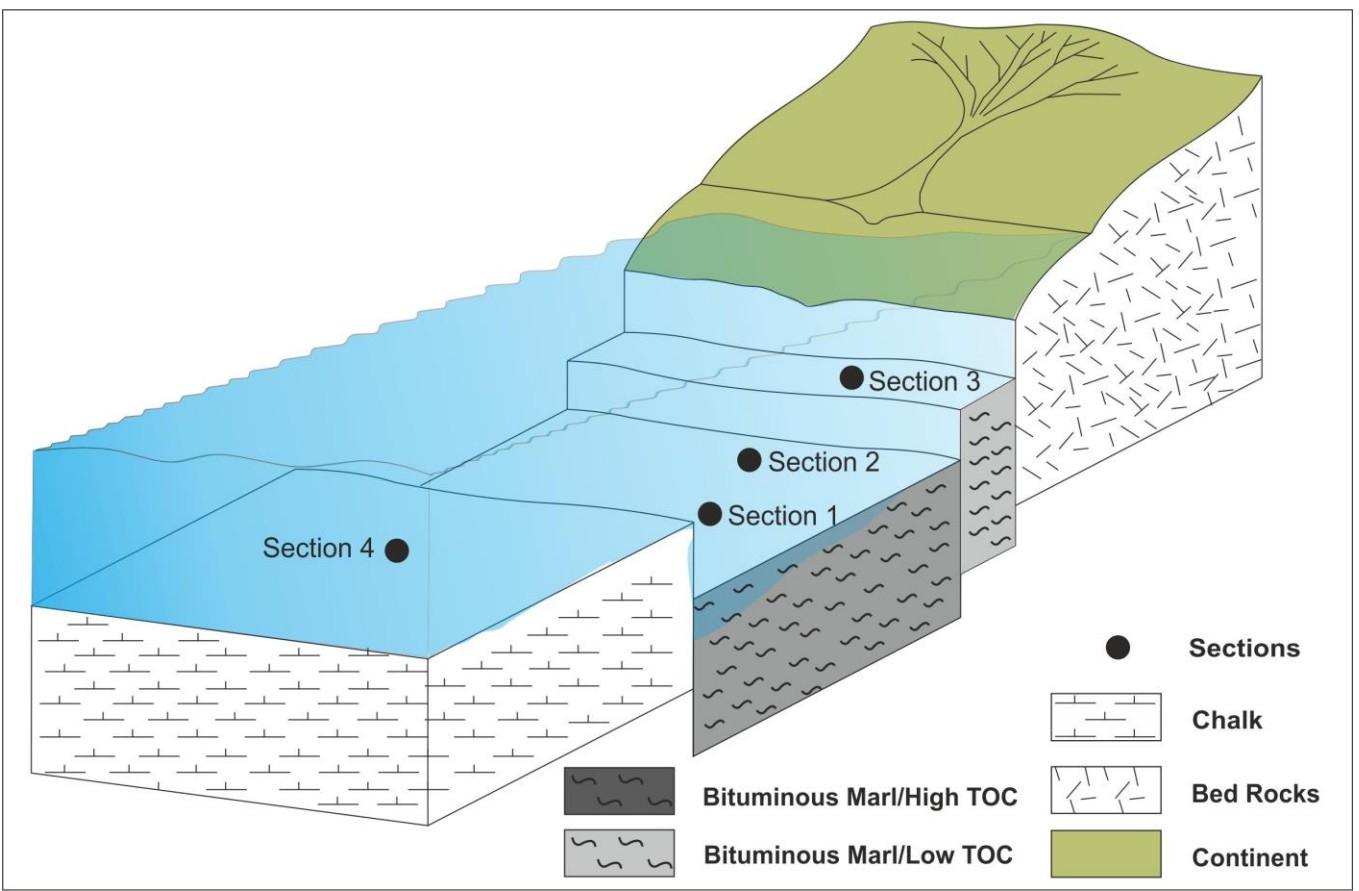

**Figure 10.** Schematic diagram represents the structural model gained from nannofossils data. Explanation of the model proposes a hydrologic pattern in the studied area.

*5.5. Basin Restriction and Organic Matter Preservation*

The study area is situated within the same sedimentation regime as the African–Arabian Plate margin. It has experienced a geological evolution resulting from the formation of the Dead Sea transform fault and the Syrian arc movements, leading to the creation of deformed subbasins, including swells and sub-basins [6,10,12,16,22,24]. These resulting depressions, or sub-basins, have consistently shown higher organic matter content and enhanced preservation. This supports the established theory that attributes the well-preserved organic matter of sediments in subsided blocks to enclosed environments with limited oxygen [14,34]. Moreover, the elevated energy of water has accelerated sedimentation rates, enabling the deposition of sediments in basins that would not typically receive them under normal conditions. During wet seasons, terrigenous inputs are carried into these basins, enriching them with nutrients that support thriving marine life, resulting in a higher number of nutrient indicator species [13,15,16]. The abundance and diversity of calcareous nannofossil assemblages notably increased during the UC20c interval. High-nutrient species proliferated in the basin, which experienced an acceleration in the hydrological cycle and an increase in terrigenous inputs [31].

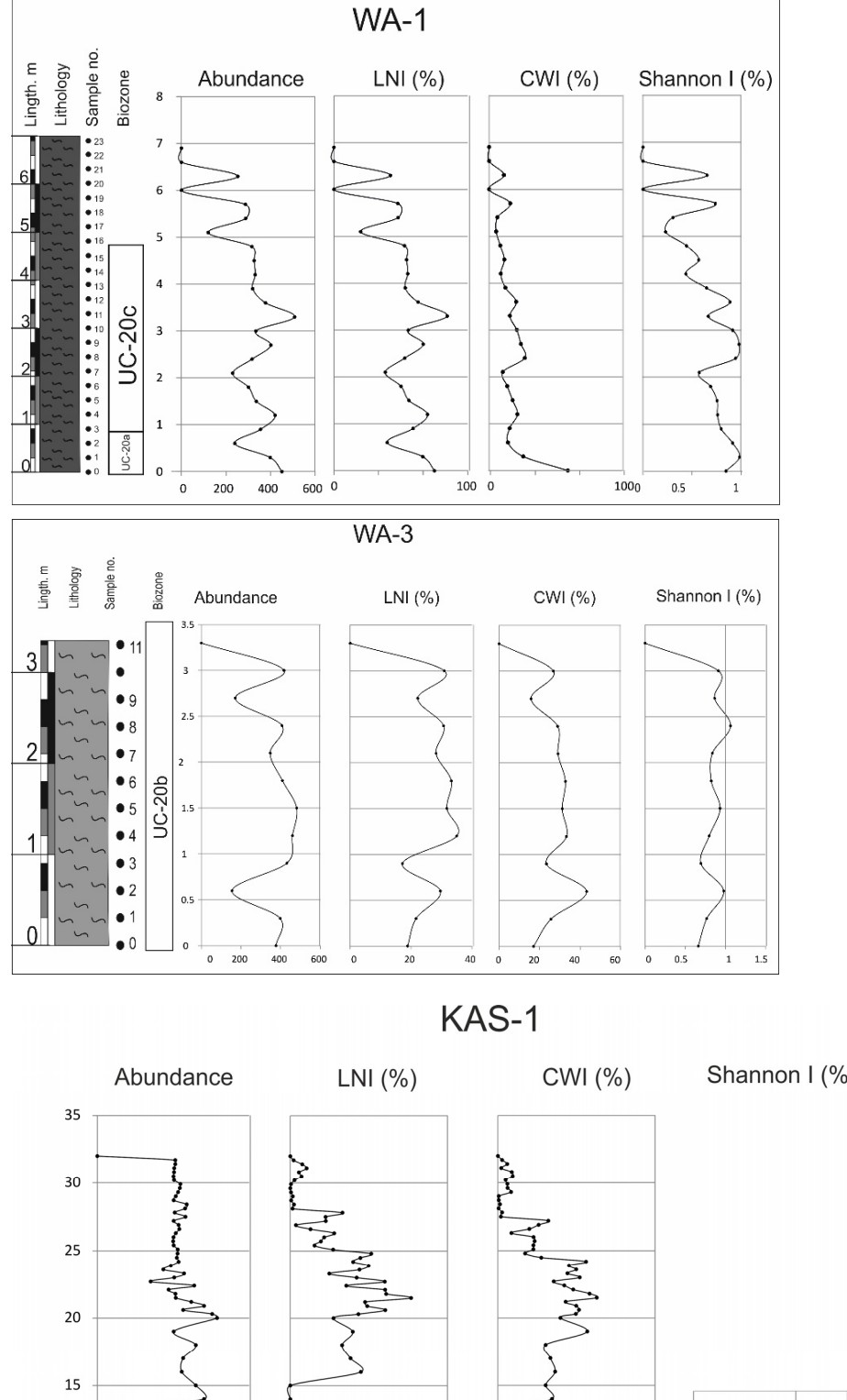

**Figure 11.** Ecological parameters from studied sections.

## 6. Conclusions

The findings derived from the analysis of calcareous nannofossil assemblages in the Wadi Arab and Kufr Asad sections yield several key observations as follows:

1. The MCM formation demonstrates a richness in calcareous nannofossils, as evidenced by the presence of numerous marker species, including *L. quadratus*, *M. murus*, *M. prinsii*, and *Cr. tenuis* in the examined sections.
2. Based on the nannofossil marker species, the age assignment of the oil shale and chalk successions (WA-1, WA-2, WA-3, KAS-1) corresponds to the Late Cretaceous–Paleogene period, delineated within the following biozones in chronological sequence: UC-20a, UC-20b, UC-20c, UC-20d, and NP-2.
3. The study identifies two hiatus intervals within section WA-1 and one within KAS-1.
4. The application of a semi-quantitative method of correlation reveals that the UC-20 interval commenced with a low sedimentation rate, subsequently intensifying due to the acceleration of topographic changes between geological blocks.
5. The UC-20C biozone reveals a clear trend toward warming and nutrient enrichment. This trend is linked to the presence of abundant and diverse species, and it aligns with wet seasons marked by an intensified continental influx into the subbasin. Consequently, nutrient levels are enhanced, facilitating the existence of diverse calcareous nannofossil assemblages.
6. During these wet periods, terrigenous inputs transported nutrients, sustaining thriving marine ecosystems. High nutrient indicator species thrived in conjunction with the abundant nutrient-rich fossils.
7. In the fourth section, a higher sampling interval is necessitated by the steep slope, posing a constraint on this research.
8. Geochemical analyses could offer valuable insights for establishing correlations with calcareous nannofossil data, particularly in cases where certain elements exhibit strong associations with terrigenous input.

**Author Contributions:** Conceptualization, A.H. and M.A.; methodology, A.H. and M.A.; formal analysis, all authors; investigation, all authors; data curation, all authors; writing—original draft preparation, O.M.A.-T.; writing—review and editing, M.A.; supervision, M.A.; funding acquisition, M.A. All authors have read and agreed to the published version of the manuscript.

**Funding:** This study is funded by the Deanship of Research and Graduate Studies of Yarmouk University; project number 65/2022.

**Institutional Review Board Statement:** Not applicable.

**Informed Consent Statement:** Not applicable.

**Data Availability Statement:** Data are contained within the article.

**Conflicts of Interest:** The authors declare no conflict of interest.

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
