# Peer review of "Calcareous Nannofossils Biostratigraphy of Late Cretaceous–Paleocene Successions from Northern Jordan and Their Implications for Basin Analysis"

_geosciences, doi:10.3390/geosciences13110351_

Round 1

Reviewer 1 Report

Comments and Suggestions for Authors

Dear Authors,

I’ve examined your manuscript with great interest. Indeed, it seems to be based on a good research project, and it is informative to certain degree. The topic will be interesting to many specialists in the world, and it is suitable to the journal. Such classical biostratigraphical studies remain very urgent in the modern geology. Nonetheless, the interpretations should be argued better, and the references need update. The manuscript seems to be written in rush, and its technical side is far from being accurate. Numerous amendments are necessary, and I hope my comments will help you to achieve this task.

1)      Title: avoid comma after “Jordan”; please, be consistent and write either “Maastrichtian-Danian” or “Late Cretaceous-Paleocene”.

2)      Key words: please, avoid the words already used in the title.

3)      Figure 1: please, make all details well visible on the insert geological map. I’m not sure that all symbols from the stratigraphical columns are explained in the legend. Change fm to Fm. What is the source of the geological map? Which geological time scale and local stratigraphical framework you follow? Please, add citations!

4)      Subsection 2.1: I’m sure you can add citations to the fresher sources! Note also that Maastrichtian is a stage, not epoch!

5)      Section 2.2: please, refer to the present version of the geological time scale (stratigraphy.org) and the general references such as this one: https://www.sciencedirect.com/book/9780128243602/geologic-time-scale-2020

6)      Figure 2: bitumenous or bituminous?

7)      Line 135: please, cite the sources where this method is explained.

8)      Figure 3: please, improve the quality and indicate all species with authors and years.

9)      Figure 4: do you really propose biozonation or you just establish the presence of the well-known biozones in the study area?

10)  Figure 5-8: in which units the relative abundance is measured? In %%?

11)  Results: this section seems to be incomplete because it presents the only part of what is explained in the methodological part of the work.

12)  Discussion: you have to consider the global sea-level changes (check the works by Bilal Haq and Kenneth Miller). You have also cite the works explaining the local and regional tectonic activity. Without doing this, you cannot make judgments of the controls of the palaeogeographical development of this area. Both regional and global factors could be responsible for this development. Figure 10 should be better argued. This section needs total update. Please, note also that those parts of this section, which refer to the explanations in the methodological section, should be moved to Results.

13)  Conclusions: please, state the limitations of this study and the perspectives for further research.

14)  The writing is clear, but it needs significant linguistic polishing. I see errors/typos in the terms! The help of any English native speaking colleague will be suitable.

15)  References: indeed, more fresh sources should be cited!

Comments on the Quality of English Language

The writing is clear, but it needs significant linguistic polishing. I see errors/typos in the terms! The help of any English native speaking colleague will be suitable.

Author Response

Dear Professor,
The response in the attachment 

Reviewer 2 Report

Comments and Suggestions for Authors

The authors require significant revision and extensive correction of the manuscript for better presentation and clarity of the study. please find the attached annotated file for details. However, some of the key suggestions and comments are as follows:

1. The similarity index is 57% which should be addressed by a major overhaul and rephrasing of the text.

Abstract section:

2. what is the target /goal of the study...define the aim and key objectives of the study in the abstract section.

3. discuss the selection of 116 samples from 4 sections? how the samples and slides were collected for further analysis.

4. it is unclear what is the comparative analysis of four sites and their findings. discuss them for the distribution of geological age from Maastrichtian to Paleogene.

Introduction section:

5. Introduction section dot integrates the association of calcareous nannofossils in basin analysis and its significance. discuss them accordingly..

study area section:

6. in the study area two sites are mentioned but in the abstract, there are 4 sites. please correct them.

 7. The abbreviation MCM Formation is not previously mentioned in the full form in the text. please discuss it.

8. Lines 100-102: unclear statement. please describe the relation between basin evolution and tectonic activity.

9. Lines 104-106 complex sentence structure. rephrase it for better understanding.

Material and methods section:

10. Figure 2: is there any shale in the lithological columns?

11. Figure 2: why does the WA3 section have a light grey color and other sections have a dark grey color?

12. Figure 3 caption: Explain the caption in detail for various sites and sections with sample selection.

13. the frequency of sample collection is different in the studied section. could it be considered as the limitation and constraint of the study?

Results section

14. Figure 4: The text within the figure is highly distorted due to the vertical stretching of the figure and the figure is ambiguous and not self-explanatory. redraw and explain the figure correctly.

15. subsection 4.2.2 does not include any update regarding the thickness, it only discusses the finding regarding the preservation.

16. Please discuss how the preservation is moderate to well. I can not understand the variation of preservation potential among the studied profile

17. Figure 5: The text written at the top of columns is too small and. increase the font size for clarity.

18. Figure 9: The font size of WA-1 and KAS-1 is very big. adjust all font sizes of figures for better visibility.

19. Figure 10: The authors have not mentioned any shale in the lithology presented in Figure 2. However, the model shows abundant shale in sections 1 and 2. Clarify it.

Reference section

20. recent literature is missing. include some key updated references in the bibliography section.

Comments on the Quality of English Language

omit complex sentence structures.

significant rephrasing is required for clarity and better understanding.

Author Response

Dear Professor, 
the response is in the attachment below.

Round 2

Reviewer 1 Report

Comments and Suggestions for Authors

Dear Authors,

I'm generally satisfied with your revisions and responses. Indeed, some polishing of the writing is still necessary, but I hope the MDPI's linguistic service will help with this. So, I recommend acceptance.

Comments on the Quality of English Language

Indeed, some polishing of the writing is still necessary, but I hope the MDPI's linguistic service will help with this at the stage of proof preparation.

Author Response

Dear Dear Editor of Geosciences, 

the response is in the attachment below0

regards

Reviewer 2 Report

Comments and Suggestions for Authors

The following revisions are suggested:

1. add the keywords; Biozones; hiatus intervals; rate of sedimentation

2. Figure 4: Font size of geological time scale stage is too large. adjust the font size in the figure.

Author Response

Dear Editor of Geosciences,

the response is in the attachment below.

regards
